# Sequence and Structure-Based Analyses of Human Ankyrin Repeats

**DOI:** 10.3390/molecules27020423

**Published:** 2022-01-10

**Authors:** Broto Chakrabarty, Nita Parekh

**Affiliations:** 1TCS Research (Life Sciences Division), Tata Consultancy Services Limited, Hyderabad 500081, India; broto.c@tcs.com; 2Centre for Computational Natural Sciences and Bioinformatics, International Institute of Information Technology, Hyderabad 500032, India

**Keywords:** ankyrin repeats, human ankyrins, protein contact network, ankyrin conserved residues, ankyrin repeat domain

## Abstract

Ankyrin is one of the most abundant protein repeat families found across all forms of life. It is found in a variety of multi-domain and single domain proteins in humans with diverse number of repeating units. They are observed to occur in several functionally diverse proteins, such as transcriptional initiators, cell cycle regulators, cytoskeletal organizers, ion transporters, signal transducers, developmental regulators, and toxins, and, consequently, defects in ankyrin repeat proteins have been associated with a number of human diseases. In this study, we have classified the human ankyrin proteins into clusters based on the sequence similarity in their ankyrin repeat domains. We analyzed the amino acid compositional bias and consensus ankyrin motif sequence of the clusters to understand the diversity of the human ankyrin proteins. We carried out network-based structural analysis of human ankyrin proteins across different clusters and showed the association of conserved residues with topologically important residues identified by network centrality measures. The analysis of conserved and structurally important residues helps in understanding their role in structural stability and function of these proteins. In this paper, we also discuss the significance of these conserved residues in disease association across the human ankyrin protein clusters.

## 1. Introduction

Ankyrin (ANK) repeat domains are among the most common structural motifs known in proteins. It is found across all phyla and with high abundance in eukaryotes [1]. Some ANK-repeat proteins consist solely of ANK repeats; others are multidomain molecules in which ANK repeats are combined with other unrelated structural modules and take part in many processes, including transcription initiation, cell cycle regulation, and cell signaling [2,3]. The ANK repeat motif is typically 30–34 amino acid residues and has helix-turn-helix conformation with short loops at the N and C termini, as shown in Figure 1a. In general, 4–6 copies of the repeat stack onto each other, forming an elongated structure mediated by the conserved hydrophobic faces of the helices [4]. The last and first two residues of adjacent repeats form a β-turn, which protrude outwards at an angle of ~90° to the antiparallel α-helices, resulting in the characteristic L-shaped cross section [5]. The 3D structure of a designed ankyrin repeat protein with 4 repeat copies is shown in Figure 1b. A left-handed twist to ankyrin repeats results in the repeat stack being slightly curved, creating concave and convex faces. Less conserved positions in the motif are located on the surface and are likely to interact with ligands, while more conserved positions are buried in the structure and are responsible for the structural stability of the domain. This is achieved by forming both intra- and inter-repeat non-bonded interactions, such as hydrophobic and hydrogen bond interactions. However, a single ankyrin repeat motif is unable to adopt a folded structure, and at least two copies of the ANK motif in tandem are required to form the hydrophobic core. Consecutive ankyrin repeats form a single, linear solenoid structure called ankyrin repeat domain, which is one of the most common protein–protein interaction platforms in nature. Many important functions of ankyrin-like proteins are mediated by protein-protein interactions provided by ankyrin repeats, such as the formation of transcription complexes, initiation of immune responses, biogenesis and assembly of cation channels in the membranes, and regulation of some cell-cycle stages. Thus, mutations in genes that encode ankyrin-like proteins can cause defects in gene expression, leading to onset and progression of many diseases [6]. Examples of disease associated ankyrin proteins include cell cycle inhibitor p16 (CDKN2A), which is associated with cancer and Notch protein, a key component of cell signaling pathways that is associated with the neurological disorder CADASIL [7,8].

The ankyrin fold appears to be defined by its structure rather than its function, since there is no specific sequence or structure that is universally recognized by it and interacting residues are discontinuously dispersed in the whole protein [9]. In addition, large variation in copy number is observed across different proteins. It would be interesting to understand the role of conserved residues in the ANK motif on its structure, function, and disease association. In this study, we carried out sequence and structure-based analysis of conserved positions in the Ankyrin motif. In this paper, we discuss some of these positions involved in disease-causing mutations in the ankyrin repeat region. We first clustered human ankyrin proteins using CD-HIT to identify proteins with similar sequence. Next, we performed sequence- and structure-based analysis of the repeat regions to understand the diversity in the human ankyrin proteins. We identified the critical residue positions in the Ankyrin motif, and we discuss their role in diseases in this paper.

## 2. Results

### 2.1. Human Ankyrins in UniProtKB

The UniProtKB database (2021_09 release) reports 965 distinct entries for human proteins containing ankyrin repeats. These include some low confidence unreviewed entries and multiple isoforms of a gene product. We extracted 257 well-annotated reviewed ankyrin repeat entries from UniProtKB database that are considered for analysis in this study. The details of these proteins are given in Appendix A. Copy number analysis in 257 human ANK proteins revealed that about one-third of these proteins contain 5–6 copies of the repeating units, as seen in the frequency distribution plot in Figure 2a. In Figure 2b, the ankyrin domain coverage with respect to the number of repeat copies present in the protein is depicted. Ankyrin domain coverage is computed as the ratio of the number of amino acid residues reported in ankyrin copies and the total number of residues in the protein. We observe that, irrespective of the number of repeat copies, most human ankyrin proteins are multi-domain, with significant proportion of the proteins being covered by the non-ankyrin region. The average coverage of the ankyrin repeat region is ~33% across all the proteins. PSMD10, ANKRD44, and ANKRD52 exhibit the maximum coverage of the ankyrin region, while ANKLE2, YTHDC2, and ANKRD12 are observed to have the least coverage (less than 5%). The frequency distribution of the length of ankyrin repeat copies across 257 human ankyrin proteins is shown in Figure 2c. We observe high consistency in the length of known ankyrins, with ~91% of the repeat copies being reported in the length range of 30–33 amino acid residues.

Sequences of all ankyrin repeat copies in 257 human ANK proteins are extracted from UniProtKB and Multiple Sequence Alignment (MSA) performed on these repeat units using Clustal Omega [10]. The sequence logo of the MSA is obtained using WebLogo3 [11] and given in Figure 2d. The highly conserved positions proposed by Kohl et al. [12] and Mosavi et al. [13] are observed with high frequency in the MSA: conserved tetrapeptide motif TPLH at positions 4–7, Glycine at positions 2, 13, and 25, Alanine at position 9 and 26, and Leucine at positions 21–22. The conserved Glycine residues at positions 2, 13, and 25 are part of the two-residue turns T_1_, T_2_, and T_3_, respectively, of the ankyrin motif, as shown in Figure 1. The first helix (H_1_) has 4 conserved positions—Proline at 5, Leucine at 6, Histidine at 7, and Alanine at position 9, while there are only 2 conserved positions in second helix (H_2_)—Leucine at positions 21 and 22. To see if there is any bias in the occurrence of amino acids in ANK repeat regions, the frequency of occurrence of 20 amino acids (in %) in repeat regions is compared with their occurrence in 565,254 protein entries in UniProtKB/Swiss-Prot protein knowledgebase release 2021_03 [14]. It may be noted from amino acid compositional bias in ankyrin repeats in Figure 2e that the composition of non-polar amino acids Leucine and Alanine, and positively charged amino acid Histidine is significantly higher in ankyrin repeat regions. These three amino acids are also among the most conserved residues found in the consensus. The compositional bias in the increase in Histidine composition is compensated by decrease in other positively charged residues Lysine and Arginine. Similarly, the bias towards the non-polar amino acids Leucine and Alanine is compensated by bias against other non-polar amino acids Isoleucine, Proline, and Phenylalanine.

### 2.2. Sequence Clusters

The ANK domain is defined by the occurrence of 3 or more consecutive copies of the repeating unit. Hence, we filtered the entries with ≥3 repeat copies for further analysis. This resulted in a total of 216 human ankyrin proteins. We used multiple iteration CD-HIT method (H-CD-HIT) [15] for clustering the human ankyrin repeat domains. Three iterations of CD-HIT were executed with identity thresholds of 90%, 60%, and 30%, respectively, to obtain clusters of ANK repeat proteins. The iterative execution of the CD-HIT method with decreasing thresholds helps in identifying distantly related sequences, as only one representative sequence from a stringent cluster is considered in the subsequent iterations of clustering. The set of 216 ANK proteins (≥3 copies) resulted in 49 clusters, of which more than half (26 clusters) had only 1–2 members. The cluster membership of the human ANK proteins and their sequence identity with cluster representative is given in Appendix A. Detailed analysis of top 10 clusters with >5 members was carried out and is summarized in Table 1. Clusters 1 and 2 are significantly large with over 30 members. Clusters 1 and 3 exhibit high standard deviation in the number of copies, indicating large variation in the copy number among their members. Clusters 2, 4, 5, 9, and 10 exhibit low standard deviation, indicating high similarity in number of copies among their members.

The domain architecture of cluster members was obtained from InterPro and is compiled for each cluster in Table 1. Except clusters 7 and 8 that are single domain proteins, all other clusters are multi-domain proteins containing one or more domains apart from the ankyrin repeat domain (ARD). Though clustering was carried out only based on ankyrin repeat domain sequence, overall domain architecture of member proteins in a cluster is found to be similar. Some members of cluster 1 contain ARD on the N-terminal region, followed by spectrin-binding ZU5 domain and death domain on the C-terminal end of the protein. The notch signaling proteins in cluster 1 contain the Notch domain and calcium-binding EGF-like domain, along with ARD. All the proteins in cluster 2 contain CCDC144C-like coiled-coil domain, along with ARD. The structure and function of cluster 2 proteins are not well-studied, with no structural data available for any protein in the PDB. Some members of cluster 3 have protein kinase domain, along with ARD, while some have the nucleic acid recognizing K homology domain. Cluster 4 also consists of the nucleic acid binding domain, BRCT domain, and PCGF1 binding domain, along with ARD. The NFkappaB proteins of cluster 5 contain 4 domains—NfkappaB IPT domain, Rel homology domain (RHD), Death domain, and ARD. All members of cluster 6 have the SOCS box domain at the C-terminal and ARD at the N-terminal, which facilitate the ubiquitination of tumour necrosis factor receptor II. The proteins in clusters 9 and 10 have the ion transport domain, along with ARD.

For each of the top 10 clusters, MSA of repeat copies of its members was carried out using Clustal Omega, and sequence logo representative of each cluster was obtained, as shown in Figure 3. To construct sequence logo, the consensus at 50% was obtained using SeaView [16] and is shown in Table 2. Although there are some similarities among the sequences of the cluster and the overall consensus shown in Figure 2d, there are distinct identifiable differences in key positions among the clusters. Alanine at position 9 and Leucine at positions 6, 21, and 22 are conserved in all the clusters. The consensus of Cluster 1, which is the largest cluster, matches very well with the overall consensus of human ankyrin in Figure 2d. Cluster 2 shows differences with the overall ankyrin consensus at many positions. It has Arginine at position 2, Alanine at position 5, Valine at position 10, and the frequency of Glycine at positions 2, 13, and 25, and Histidine at 7 and 14 is significantly reduced. Cluster 3 is very similar to the overall consensus and Cluster 1, except that the Histidine residue is not conserved at position 7. Clusters 4, 6, 7, and 8 show high similarity with overall consensus on all well-conserved positions; however, variation in the level of conservation of these positions was observed. Cluster 5 is very well conserved in all the positions, except the Glycine and Histidine at positions 13 and 14, respectively, that are part of the turn T_2_ between the two helices. In cluster 9, the residues of the first helix are well-conserved, but the residues in the intermediate turn, second helix, and the long loop are not conserved. Cluster 10 exhibits maximum deviation from the consensus, especially in the first helix region. Except for Alanine at position 9 and Leucine at 21–22, other positions exhibit high variability for cluster 10.

### 2.3. Structure Analysis

Structural analysis of 127 ANK repeat proteins in the top 10 clusters was carried out. Of these, only for 23 proteins structural information is reported for the repeat region in Protein Data Bank [17]. For 102 of the remaining 104 proteins, predicted structure is obtained from AlphaFold DB [18]. We extracted structural coordinates for residues in repeat domain and considered 79 structures that had very high model confidence score. Thus, the structural ANK repeat dataset considered for the analysis consists of 102 structures (23 PDB structures and 79 high confidence AlphaFold predicted structures) and their details is summarized in Appendix A. We analyzed the secondary structure architecture (using DSSP [19]) and topological properties derived from the network representation of these protein structures (using NAPS [20]).

#### 2.3.1. Secondary Structure Architecture

For each cluster, secondary structure assignment information is extracted for all repeat copies of member proteins using DSSP program [19], and the consensus secondary structure for the cluster is obtained. Secondary structure assignment with highest frequency at each position in the repeat copy is considered for building the consensus secondary structure for each cluster and is given in Figure 4. As expected, the Helix-Turn Helix motif is well conserved across the 10 clusters, with the two helices of lengths 7 and 9 amino acid residues from positions 5 to 11 and 15 to 23, respectively, with minor variations in clusters 3 (first helix longer by a residue) and 9 (second helix longer by a residue). Four turn positions of two-residues each (1, 12, 24, and 28) are also well conserved, with slight variations in clusters 2, 3, 9, and 10.

#### 2.3.2. Network Analysis of Protein Structures

Network representation of protein structures, called protein contact networks (PCN) or residue interaction networks (RIN), have been widely used to analyze topological properties of proteins [21,22]. Concepts of graph theory have been applied for the analysis of protein structures and their function, such as identification of residues critical for protein stability and enzymatic activity, understanding allosteric regulation, identification of repeats and domains, etc. [23,24,25,26,27]. In this study, we carried out network-based analysis of ankyrin repeat domains in top 10 clusters to understand the topological importance of amino acid residues in different clusters. The Cα networks were constructed for ankyrin repeat domain regions (based on UniProtKB annotation) using the NAPS portal [20]. The average residue-wise connectivity information for the 10 clusters is depicted in Figure 5 by overlaying line plots of each cluster. The degree plot in Figure 5a represents the total number of interactions of the node. It is observed that residues 9 and 10 in the first helix have the maximum degree. We further analyzed the interaction patterns by segregating the inter- and intra-repeat edges, as shown in Figure 5b,c respectively. The inter- and intra-repeat edges for an example designed ankyrin repeat copy with its adjacent copies are shown in Figure 6. The intra-repeat edges are shown in grey color, and the inter-repeat edges are shown in yellow. The intra-repeat unit edges capture the non-bonded interactions within the secondary structure elements and those between secondary structure elements and are responsible in stabilizing the structural motif. Residue 9 has the highest degree in all the clusters and has Alanine conserved across all the clusters. It has high intra-repeat edges, while residue 10 has high inter-repeat edges. The residues at positions 5, 6, 9, and 10 of first helix interact with the residues 17, 18, 21, and 22 of the second helix, which is reflected in the high intra-repeat edges for these residues [5]. These include the conserved residues Pro5, Leu6, Ala9, Leu21, and Leu22. For the region towards the end of second helix and the long loop at the C-terminal end, the number of intra-repeat edges are highly conserved across all clusters, except in clusters 9 and 10. The inter-repeat unit edges capture non-bonded interactions with residues of adjacent repeat copies and dictate the overall fold of the domain. The N- and C-terminal residues, namely 1, 2, 29, and 30, show high inter-residue interactions, which is expected due to the proximity with the adjacent repeat copies in the primary structure. Among these terminal residues, Glycine is highly conserved at position 2 in most clusters, and Asparagine or Aspartic acid is observed at position 29. The residues in the first helix are involved in inter-repeat interactions, whereas, except residue 17, the other residues of the second helix are not involved in inter-repeat interactions. The inter-repeat interaction pattern is much diverse across the clusters as compared to the intra-repeat interaction pattern. Lower conservation at the sequence level in clusters 9 and 10 (Figure 3) is also reflected in their poor overlap with other profiles. The highly conserved residues across all the clusters in Figure 3, such as L at position 6, 21, and 22, and A at position 9, exhibit conserved inter-repeat interactions. These observations suggest that the interaction pattern within individual repeat copies is highly conserved across all human ankyrin proteins, while the functional diversity among the proteins can be attributed to the variation in the inter-repeat interactions between them. The inter- and intra-repeat interactions also dictate the 3D fold of the ankyrin motif.

Two other node centrality measures, namely betweenness and eigenvector centrality profiles of the repeat motifs, are analyzed in top 10 clusters to identify topologically important residues. In structural repeat proteins, the centrality values of the central repeat copy are higher in magnitude and taper down for neighboring copies on either side, though the general trend is the same for all copies [28]. Hence, the centrality value of each repeat unit is normalized with respect to the maximum value in the respective unit, so that the centrality values for each copy lies in the range 0–1. Next, the consensus profile for each cluster is constructed by taking the average centrality value at each position in the motif across all repeat copies of all members of the cluster. The consensus or representative betweenness and eigenvector centrality profiles for the top 10 clusters is shown in Figure 7a,b, respectively. The betweenness centrality is the shortest-path based node centrality measure that captures the importance of a node in transmission flow within the network [29]. In the network representation of protein structures, amino acid residues with high betweenness centrality have been found critical for folding of the protein [30]. From Figure 7a, we observe that residues of the first helix of the ankyrin motif exhibits higher betweenness centrality values compared to those of the second helix. This is because the first helix is buried in the core of the protein. Residues positions 10 and 17 are observed to have higher number of inter-repeat interactions and also exhibit higher betweenness centrality values. This is not surprising as these residues act as linking residues between the adjacent repeat copies, and most of the shortest paths between the repeat copies pass through these nodes.

The sequence, structure, and functional domain information analyses provided in this study help in functional grouping of human ankyrin proteins. Protein clusters identified based on the ankyrin repeat domain sequence show a distinct conservation pattern at certain key positions in the ankyrin motif. Further, network-based structural analysis helps in identifying conserved interaction patterns across ankyrin protein clusters. Below, we discuss the impact of variations at these conserved positions and disease association in few representative examples of human ankyrin proteins across the clusters.

The principal eigenvector component of the adjacency matrix, called eigenvector centrality, captures not only the information of direct connections of a node (degree) but also the connectivity information of its neighbors, neighbor’s neighbors, and so on [31,32]. In our earlier work, we have shown that different protein repeat families exhibit a highly conserved and distinct eigenvector centrality profile [28]. We showed that these conserved patterns can be used to identify structural repeats in proteins [33,34] and developed a database of structural repeat proteins, called DbStRiPs [35]. The average eigenvector centrality profile is highly conserved across all the 10 clusters, as shown in Figure 7b, in accordance with the profile used in DbStRiPs. The amino acid residues of the first helix that are towards the core of the protein and the residues 17 and 18 from second helix that have high inter- and intra-repeat unit interactions, respectively, also have high eigenvector centrality. Although the sequence conservation between the repeat units is poor, and the domain architecture of the clusters is different, the conserved eigenvector centrality shows that the overall connectivity of the residues within the repeating units remain highly conserved, which is the characteristic feature of Class III and Class IV repeats in Kajava’s classification [36].

## 3. Discussion

### 3.1. Variations in ANK Structure

Though the topology of the structural ANK motif is well-conserved, some differences are observed across different ankyrin repeat proteins. One of the major factors responsible for these differences is the presence of non-ankyrin domains that affect the orientation of the ANK domain in the overall 3D structure of the protein. The 3D protein structures for representative example proteins from top 4 clusters are shown in Figure 8, with the ankyrin repeat domain depicted in ‘red’, while all other domains are shown in ‘grey’. In these multi-domain proteins, the residues of ankyrin repeat domain have long-range non-bonded interactions with the residues of the other domains. The ankyrin repeat domain of ANK1 protein in Figure 8a forms a super-helical structure around the other domains. The ankyrin repeat domain of protein POTEJ (Figure 8b) has interactions with other domains through the convex surface residues, while that in TNI3K (Figure 8c) interacts with other domains through its concave surface residues. In the case of protein BARD1, the interaction of the ankyrin repeat domain with other domains is through the residues of its surface orthogonal to the concave and convex surfaces, as shown in Figure 8d. These differences in the inter-domain interaction patterns between the clusters affect the orientation of the ANK domain. The other factor affecting the topology of the ANK motifs is the presence of number of contiguous ANK units in a protein. Proteins with higher number of repeats typically exhibit a more compact and concave surface of the ankyrin domain. Third, due to higher solvent accessibility of the N- and C-terminal motifs, these exhibit more flexibility compared to internal ANK motifs and may sometimes be partially truncated. Apart from these factors, evolutionary processes, such as insertions and deletions, result in variable length of the helices and the loop regions, though the overall helix-turn-helix topology maybe conserved. These variations in ankyrin repeat proteins may be responsible for the structural differences observed across different clusters and result in differences in the edge distributions of certain positions. For example, high consistency in the intra-repeat edges among the clusters, and, for some key positions, such as position 2 of the first turn, T_1_, and positions 7–10 of first helix, H_1_, high diversity in the inter-repeat edges is observed.

### 3.2. Disease Associated Variations

From multiple sequence alignment of ankyrin repeat sequences in Figure 2d, we observe patterns of conservedness in ANK motifs, for example, Glycine residues conserved at positions 2, 13, and 25, the TPLH motif at positions 4–7, Alanine at positions 9 and 26, and Leucine at positions 21 and 22. These positions have been reported to be evolutionarily conserved, indicating their importance [5]. Of the three Glycine positions, Gly13 and Gly25 terminate the two helices. The tetrapeptide motif, Thr-Pro-Leu-His (TPLH), initiates the first α-helix and forms a turn that is stabilized through reciprocal hydrogen-bonding interactions between the side-chain and main-chain atoms of Threonine and Histidine and H-bond between Proline and Histidine (H), and threonine (T) initiates the characteristic L form of the ANK repeat [5]. The conserved hydrophobic residues of the two anti-parallel helices are involved in stacking of the repeats and result in a very stable, nonglobular ankyrin domain structure with a hydrophobic core. A conserved motif GADVN is observed at position 25–29 in the loop region connecting the second α-helix with β-turn. In accordance with the analysis of Utges et al., here, Ala26 and Val28 form intra- and inter-repeat interactions, whereas Asp27 and Asn29 form hydrogen bonds, with adjacent repeats. This region spanning from GADVN to β-turn at the interface between adjacent repeats is shown to be the most flexible region of the motif through molecular dynamics simulations [37]. At positions 17–22 in the second α-helix, a hydrophobic motif, [I/V]VXLLL, is observed, with X usually being hydrophilic. These residues form intra- and inter-repeat hydrophobic networks that help to keep together the ankyrin repeat domain (ARD) structure.

The conserved positions are identified to be involved in the fold and stability of individual repeating unit or in forming the interface between repeats. On the other hand, positions 3, 12, 23, and 24 in the ANK motif are observed to be highly variable and may be considered tolerant to mutations. It would be interesting to assess the distribution of disease-causing mutations across the conserved and variable positions. As shown in Figure 1b, for a designed ankyrin protein, the ANK domain forms two surfaces: concave (coloured in ‘red’) and convex (in ‘blue’). The concave surface comprises the β-turn/loop region and the first α-helix and is often associated with protein-protein interactions (PPIs), namely the residues 1, 2, 3, 8, 11, 12, 32, 33 (32–12), and 7. Most of these positions on the concave surface are highly diverse positions, and this variability allows it to bind to different substrates. The residues on the convex surface, namely 13, 14, 15, 16, 19, 20, 22, 23, and 24 (13–31). These include the conserved positions Gly13, the conserved motifs, [I/V]VXLLL at position 17–22, and GADVN at position 25–29, which include the conserved positions Gly25, Leu21, Leu22, and Ala26. Positions 13 and 14, which define the start of convex surface, are also enriched in PPIs, probably because of their close proximity to the concave surface. Thus, less conserved positions of the ANK motif are involved in ligand interactions, while most conserved positions with high intra- and inter-repeat contacts are responsible for the stability of the 3Dfold of the ankyrin domain. Below, the variations reported at conserved positions in the ANK motif are discussed for a few proteins based on the analyses of sequence, structure, and diseases associated with these positions.

#### 3.2.1. Variations at Conserved TPLH Motif

One of the most conserved regions in the ANK repeat is the tetrapeptide motif TPLH, or its close variant TP/ALH, at positions 4–7, and is observed in most of the clusters in our analysis. It occurs at the start of first helix (H_1_) and contributes to the conformational stability locally through H-bonding network, mainly for the intermediate copies, since the terminal copies tend to exhibit more sequence diversity [38]. Proline at the 5th position initiates the helix, while Leucine at the 6th position forms multiple hydrophobic interactions within the repeat, as shown in Figure 5b and Figure 6a. Threonine at position 4 forms three H-bonds with Histidine at the 7th position, and their side chains are involved in several intra-repeat, as well as inter-repeat, hydrogen bonds. Thus, high conservation of this motif in ankyrin repeat proteins emphasizes the importance of this inter-repeat H-bond network, propagating over the entire domain. The motif is conserved across all the clusters, except cluster 10, which also shows diverse interaction pattern and edge distribution. Mutagenesis analysis of TPLH motif in the Gankyrin protein carried out by Guo et al. [38] revealed that the TPLH motif in the third and fourth ANK copies contributed significantly to conformational stability, while, in other copies (repeat copies 1, 2, 5, and 6), it exhibited comparatively lesser, or even negative, contributions.

Mutations in the repeat region of CDKN2A have been found to be associated with multiple cancers. Different sub-types of melanomas are associated with mutations in the well-conserved first helix residues. Mutations of Pro to Leu, Thr, and Ser at position 5 lead to the loss of its binding with CDK4, which is associated with multiple sub-types of melanoma, non-small cell lung carcinoma, and head and neck tumors, as shown in Table 3. Mutation of Leu to Pro and Arg at position 6 are also associated with melanoma, while mutations in position 7 from His to Asn and Tyr are reported in lung and pancreas tumors, respectively. Mutation of Leu to Arg at position 6 in ANK1 protein is associated with Spherocytosis, while Leu to Ser in INVS impairs its ability to target DVL1 for degradation.

#### 3.2.2. Variations at Conserved Glycine Positions 

The Glycine at positions 2, 13, and 25 are part of the two-residue turns T_1_, T_2_, and T_3_, respectively (Figure 1a), and observed to be well-conserved. Among the top 10 clusters, Gly is well conserved at position 2 for all the clusters, except cluster 10, while it is conserved at positions 13 and 25 in 6 out of 10 clusters. Gly2 is conserved at the beta turn T_1_ and is observed to exhibit high inter-repeat interactions in Figure 5c. Gly13 is found in the loop (turn T_2_) between the two anti-parallel helices. Since it is in close proximity to the concave surface residues, Gly13 is also enriched in PPIs. Gly25 breaks the second helix. The Gly residues in all the positions are part of a low number of interactions due to its small size, and, consequently, lower values of network centrality are observed. However, Gly2 exhibits the maximum number of inter-repeat edges in all the clusters due its proximity with the residues of the adjacent repeat copy. In Table 3, the number of proteins containing mutations at the three conserved positions, 2, 13, and 25, is given. Given the importance of these positions in the conformational stability of the ankyrin repeat domain, it is not surprising that only three variations at Gly2 position, three at Gly13, and six at Gly25 position have been reported. These include three cyclin-dependent kinase inhibitors, CDKN2A, CDKN2B, and CDKN2C, as well as nuclear factor kappa B1 (NFKB1) transcription factor, and ANK1 and ANKK1 genes, which have been associated with tumorigenesis.

#### 3.2.3. Variations at Conserved Alanine Positions

Alanine is highly conserved at position 9 for all the top 10 clusters, while it is weakly conserved at position 26 in 8 clusters. Ala9 is part of the first helix and has maximum interactions compared to all other positions in the ankyrin motif. The central location of the residue reflects in the high eigenvector centrality of the residue. It is involved in intra-repeat interactions and contributes to the structural stability of the ankyrin motif. Mutation of Ala to Thr at position 9 in CDKN2A leads to loss of binding with CDK4, which is responsible for CMM2 (Table 3). Ala26 is part of the long loop towards the end of the repeating unit and is part of the flexible region of the domain, which is reflected in the low number of interactions and network centrality values for the residue.

#### 3.2.4. Variations at Conserved Leucine Positions

Apart from Leu6, which is part of the TPLH motif discussed earlier, Leu is conserved at positions 21 and 22, which are part of the second helix. These are highly conserved in all the top 10 clusters and play a vital role in stabilizing the helix-turn-helix motif. This is reflected in the high number of intra-repeat edges. Mutations in both the positions in CDKN2A have been found to be associated with melanoma, as shown in Table 3.

*RNase L*: Ribonuclease L (*RNASEL*) is a key enzyme in the interferon induced antiviral and anti-proliferate pathway and is implicated in the onset and development of viral induced cancers. It consists of three domains, the N-terminal ANK domain, the protein kinase homology domain, and the C-terminal ribonuclease domain. It was first shown in Rnase L that an ankyrin repeat region interacts with nucleic acids other than proteins, and repeats 2 and 4 are involved in the binding [39]. In UniProt, variation in this protein is reported at positions 2, 3, 7, and 18. From Figure 3, we observe that, for cluster 3, position 2 has Glycine conserved, while positions 3 and 7 are highly variable, and position 18 has hydrophobic residues, mainly Valine/Alanine. The G2S mutation is a missense variant but is likely benign. Though mutations in RNase L have been implicated in various cancers, only mutations at position 3 (in repeats 7 and 8) have been shown with reduced 2–5 A binding activity and complete loss of binding activity, when mutations are presented at both the 240 and 274 position in the protein.

*Cyclin-dependent kinase inhibitors*: The family of cyclin-dependent kinase inhibitor genes, CDKN2A (p16-INK4), CDKN2B (p15-INK4b), CDKN2C (p18-INK4c), and CDKN2D (p19-INK4d), are tumor suppressor genes generally inactivated in human cancers. The ANK repeat domains in these proteins are involved in binding to CDKs. Both CDKN2A and CDKN2B are part of the same cluster 7, while CDKN2C and CDKN2D belong to clusters 19 and 15, respectively. CDKN2A is one of the extensively studied tumor suppressor that plays a crucial role in cell cycle progression, differentiation, senescence, and apoptosis. In CDKN2A, variations have been reported for most positions of the ANK motif, and the majority of these variations have been reported to be associated with some type of cancer (Appendix A). Only at positions 2, 20, 27, and 30 have no variations been reported. Of these, apart from position 2, all other are highly variable positions and not important, either for structural stability or in PPIs. In CDKN2B, the first two ankyrin repeats mediate majority of the interactions with the CDKs. Mutations at two conserved positions in the ANK motif, position 2 (G47E) and position 5 (A50V), in the TPLH motif, have been reported. Both of these mutations are in the second ANK repeat and have been implicated in lung adenocarcinoma. It belongs to cluster 7, for which position 2 is conserved with Gly, and position 5 with Pro/Ala, as shown in Figure 3. In CDKN2C, two sites, 4 and 25, have reported variants. The A72P at position 4 (TPLH motif) in ANK copy 3 is reported in breast cancer, and loss of CDK6 interaction is reported. A variant T126M is reported at another conserved site, Gly25.

*ANKRD29*: It is an ankyrin repeat domain protein consisting of 8 copies of ANK repeat. In UniProt, two mutations are reported. One mutation is at the conserved Gly2 position in repeat 4, G112E, but no disease association or functional loss of the protein is reported. The other variation is reported at position 18, V95M, which is a somatic mutation associated with breast cancer patient. It belongs to cluster 1, which exhibits predominantly Gly in position 2. Residue position 18 is part of the second helix and exhibits high inter- and intra-repeat unit interactions. The position also exhibits high eigenvector centrality, indicating the topological importance of this position.

*TPRC6*. Six members of human transient receptor potential canonical (TRPC) proteins have been reported in UniProt with ANK repeats, of which only for TRPC6 variation at Gly13 position is reported. A G109S variation in first ANK repeat and R175Q in the third ANK repeat is reported to increase calcium ion transport and associated with familial atypical multiple mole melanoma syndrome (FSGS2). Apart from these, few other variations at 1, 12, 13, 14, 16, 26, and 29 have been reported for TRPC6 (Appendix A). For most of these positions, increased cation channel activity is observed, except for N125S, which reports decrease in calcium ion activity. However, no significant indication of channel inactivation was noted. All of the six TRPCs belong to Cluster 10, for which these positions are highly variable in Figure 3. Centrality measure analysis indicate fewer inter- and intra-edges at these positions for this cluster in Figure 5, and lower values of betweenness and eigenvector centrality are observed in Figure 7.

*ANKK1*. The ankyrin repeat and protein kinase domain-containing protein 1 (ANKK1) is a member of the Ser/Thr kinase family and involved in signal transduction pathways. They play an important role in cell proliferation, differentiation, and activate transcription factors. Two variations, S670G in ANK repeat 10 and R736L in repeat 12 has been reported for Gly13 position in ANKK1, with R736L being a somatic mutation associated with lung squamous cell carcinoma. ANKK1 belongs to cluster 1, where position 13 has Gly as the dominant amino acid. Topological analysis revealed high inter-repeat interactions for this position. Variations on all the 4 positions of the TPLH motif have been observed, as well as at positions 16, 23, 25, 27, and 29 (Appendix A). Of these, only Q717L, corresponding to position 27 of the ANK motif, is a somatic mutation associated with lung large cell carcinoma. From sequence and structure analysis, these positions are highly variable suggesting probable role in PPI, namely 13, 16, and 23.

## 4. Materials and Methods

### 4.1. Dataset Preparation

The dataset of human ankyrin repeats considered in this work was obtained by filtering the reviewed entries in UniProtKB database (2021_09 release) [14]. The domain information was obtained from InterPro database (release 87.0) [40]. The disease association of mutations in human ankyrin repeat was obtained from UniProtKB database and ClinVar database [41]. The variations reported in ClinVar were filtered by considering only the pathogenic variations that lead to change in the amino acid sequence. We considered only the reviewed entries with evidence provided by multiple submitters without any conflict. The structural data was obtained from the Protein Data Bank (PDB) [17] (if available) and AlphaFold DB [18]. The 3D protein structures in AlphaFold DB were predicted structures obtained using a deep-learning based artificial intelligence program called AlphaFold [42]. We extracted the structural coordinates for the residues in repeat domain and considered only those predicted structures that have very high model confidence score. The AlphaFold program provides residue-wise confidence score of the prediction in the range between 0 and 100, called predicted local-distance difference test (pLDDT). We considered only those structures that have the average pLDDT score ≥ 90.

### 4.2. Sequence Analysis

#### 4.2.1. Clustering

The clustering of human ankyrin repeat domains is performed using three iterations of CD-HIT algorithm [15]. It works on the principle of matching identical short substrings between sequences without an actual sequence alignment. Similarity between two sequences is computed by the overlap in the frequency of short substrings, called words, such as dipeptides, tripeptides, and so on. The relationship between overlapping word count and sequence similarity has been derived from analytical and large-scale statistical analyses. In the multiple iteration CD-HIT program, called H-CD-HIT, clustering is carried out iteratively by reducing the sequence identity threshold. In each iteration, representative sequence for each cluster is identified as the one with maximum similarity with all the members. Subsequent iterations are carried out by considering cluster representatives instead of all the sequences. This increases the reachability of the clusters in datasets of diverse sequences. The sequence identity thresholds used for the three iterations are 90%, 60%, and 30%.

#### 4.2.2. Sequence Alignment 

The multiple sequence alignment (MSA) was performed using Clustal Omega [10]. Sequence logos for the MSAs were obtained using WebLogo3 [11] and consensus sequences for the clusters obtained using SeaView [16] at 50% threshold.

### 4.3. Protein Contact Network

#### 4.3.1. Network Construction 

The 3D protein structure is represented as a Cα network by considering the backbone Cα carbon atoms of amino acid residues as nodes, and an edge is drawn between them if the Euclidean distance between them is ≤*R_c_*. We constructed unweighted Cα networks for the protein structures using the NAPS portal [20] by using the distance threshold of 7 Å (*R_c_*) and minimum residue separation threshold of 1. The connectivity information of amino acid residues is represented by a symmetric *n* × *n* adjacency matrix *A*, whose elements *A_ij_* = 1 when the residues *i* and *j* are connected; otherwise, *A_ij_* = 0.

#### 4.3.2. Betweenness Centrality 

The betweenness centrality of a node is defined as the ratio of all the shortest paths passing through a node and the total number of shortest paths in the network [29] and is given by:(1)$$B\left(u\right)=\underset{s\neq u\in V}{\sum }\underset{t\neq u\in V}{\sum }\left(\frac{\sigma_{st}\left(u\right)}{\sigma_{st}}\right)$$
where *V* is the set of all vertices in the network, *σ_st_* is the shortest path between nodes *s*, and *t* and *σ_st_*(*u*) is the shortest path between nodes *s* and *t* passing through node *u*.

#### 4.3.3. Eigenvector Centrality

The principal eigen spectra of the adjacency matrix is known to depict the topology of the graph, as apart from the connectivity of a node; it also captures the connectivity of its neighbors and their neighbors, and so on [31,32]. It is given by:(2)$$x_{i}=\frac{1}{\lambda }\overset{N}{\underset{j=1}{\sum }}A_{ij}x_{j}$$
where *A_ij_* is the *ij*th element of Adjacency matrix, *λ* is the largest eigen value of *A*, and *x_i_* is the eigenvector centrality of node *i*.

## 5. Conclusions

The ankyrin repeats are found in a large number of human proteins and exhibit variations in the number of repeat units, domain architecture, function, and consensus repeat sequence. We considered 216 well-annotated reviewed human ankyrin proteins from UniProtKB and clustered them based on sequence conservation in the ankyrin repeat domain. Compositional bias is observed in the occurrence of non-polar amino acids with Leucine and Alanine, occurring with higher abundance (~26%) in the ankyrin repeat domain. These two amino acids are also observed in the three most conserved positions across all the clusters, Leucine at positions 6 and 21, and Alanine at position 9. Structural analysis shows that the intra-repeat interactions that define the ankyrin motif are highly conserved across all the clusters, while variation in the conserved positions impact the inter-repeat unit interactions that determine the tertiary fold and function of the protein. The network-based topological analyses further demonstrate the importance of the conserved positions in the structural stability and function of the protein. This helps in understanding the association of mutations in highly conserved regions to different diseases, such as spherocytosis, hereditary cancer predisposing syndrome, melanoma, lung adenocarcinoma, etc.

## Figures and Tables

**Figure 1 molecules-27-00423-f001:**
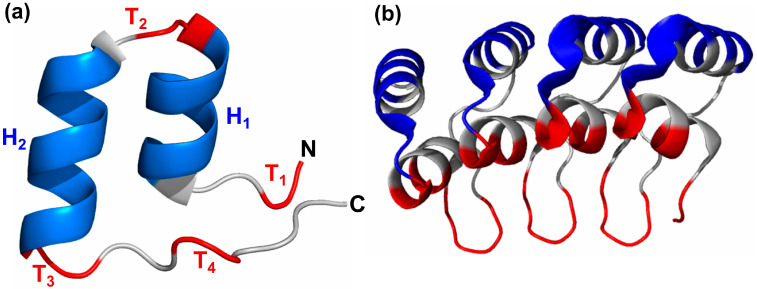
(**a**) The 3D view of Ankyrin repeat motif. The helices are colored blue, turns are red, and the coils are grey. (**b**) The Ankyrin repeat domain in designed Ankyrin repeat protein, 1N0R, comprising 4 repeat copies. The concave surface is highlighted in red, and the convex surface is highlighted in blue color.

**Figure 2 molecules-27-00423-f002:**
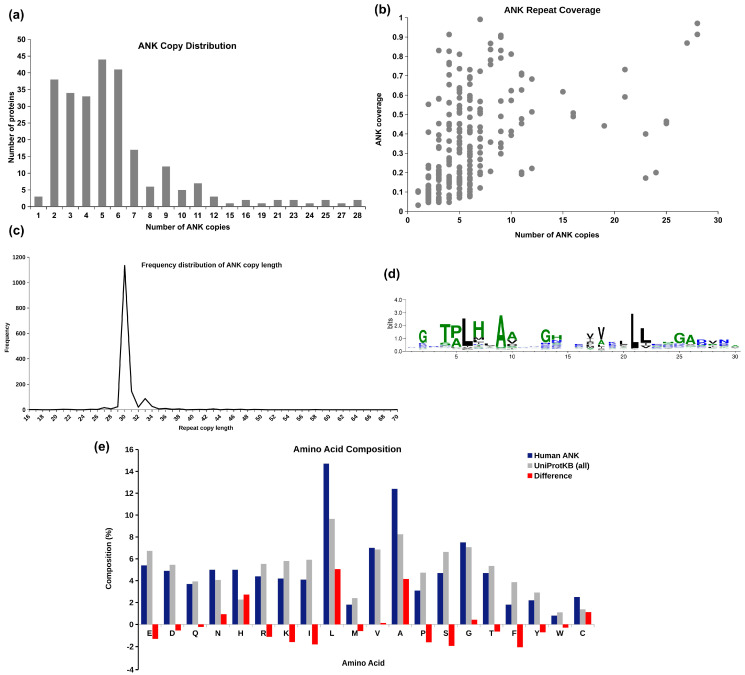
Analysis of 257 human ankyrin repeat proteins in UniProtKB. (**a**) Frequency distribution of the number of ankyrin repeat copies reported in a protein. (**b**) Coverage of ankyrin repeat region compared to the number of repeat copies. (**c**) Frequency distribution of the length of ankyrin repeat unit. (**d**) Sequence logo obtained from multiple sequence alignment of all ankyrin repeat copies. (**e**) Amino acid compositional bias in ankyrin repeat region compared to the base composition in all UniProtKB proteins.

**Figure 3 molecules-27-00423-f003:**
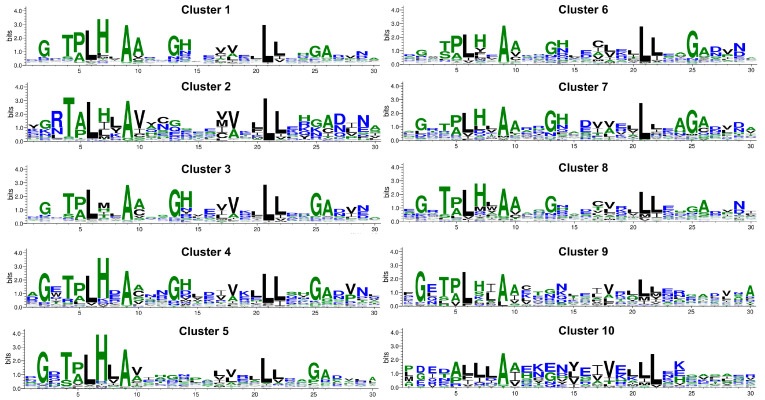
Sequence logo for the ANK motif obtained from multiple sequence alignment (MSA) of top 10 sequence clusters. The *x*-axis corresponds to the residue number in the ANK motif.

**Figure 4 molecules-27-00423-f004:**
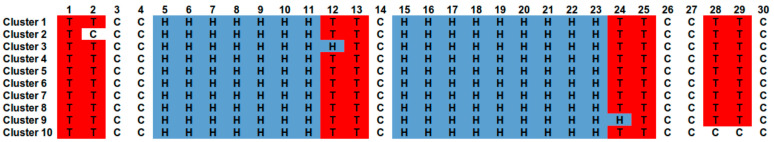
Consensus secondary structure for the top 10 clusters. H: helix, T: Turn, C: Coil.

**Figure 5 molecules-27-00423-f005:**
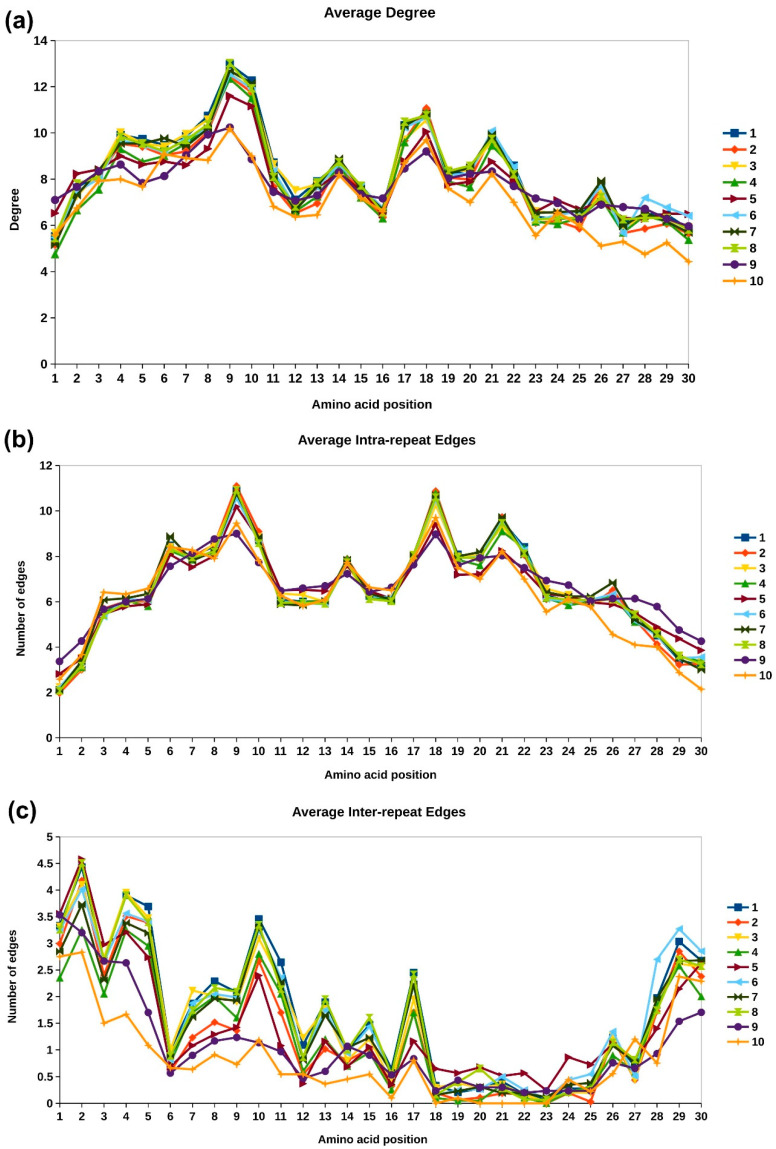
Amino acid connectivity information of the top 10 clusters. (**a**) Average number of connections (degree) for each amino acid positions within the repeat copy. (**b**) Average number of intra-repeat edges. (**c**) Average number of inter-repeat edges.

**Figure 6 molecules-27-00423-f006:**
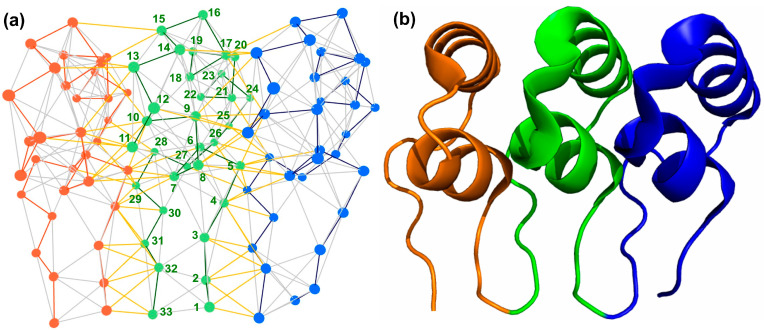
(**a**) Three-dimensional network view showing the intra-repeat edges in grey and inter-repeat edges in yellow for 3 copies of ankyrin repeat. The nodes and the backbone edges for the three copies are shown in blue, green, and orange color, respectively. The residue positions in the intermediate copy (green) are marked. (**b**) The 3D structure view of the 3 copies is shown in the same orientation as shown in the network view.

**Figure 7 molecules-27-00423-f007:**
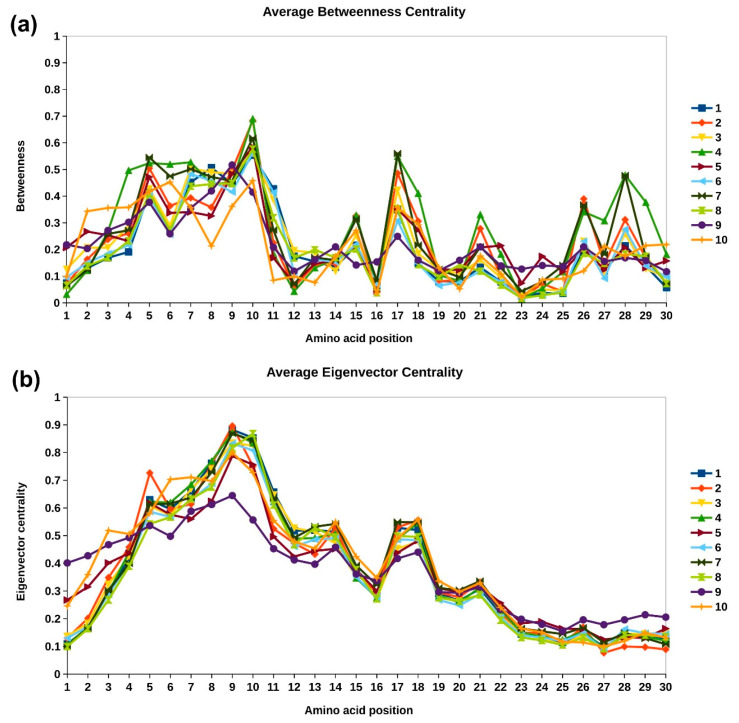
Average node centrality profiles for top 10 clusters. (**a**) Betweenness centrality. (**b**) Eigenvector centrality.

**Figure 8 molecules-27-00423-f008:**
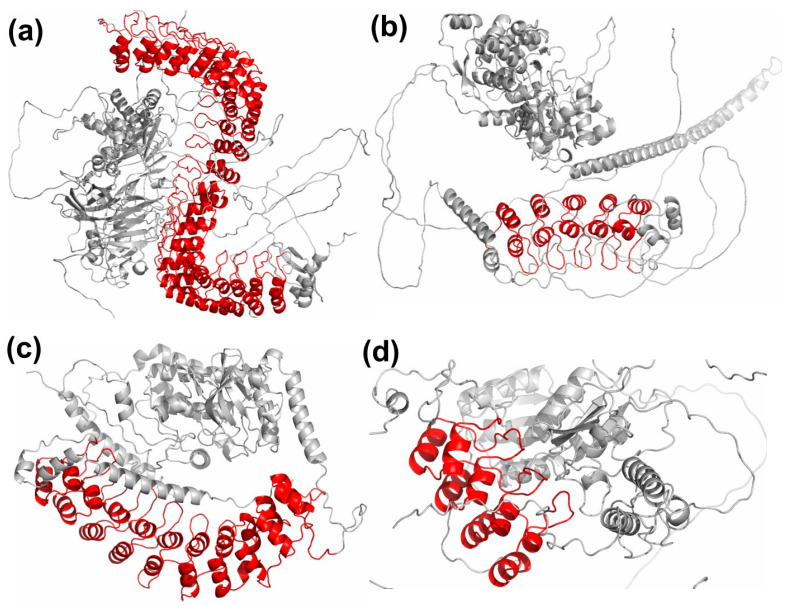
The 3D view of example ankyrin repeat proteins from top 4 clusters are shown. The ankyrin repeat domain is colored in red, and all other domains are shown in grey. The 3D coordinates of the proteins are obtained from AlphaFold DB. (**a**) ANK1 (UniProt: P16157) from Cluster 1. (**b**) POTEJ (UniProt: P0CG39) from Cluster 2. (**c**) TNI3K (UniProt: Q59H18) from Cluster 3. (**d**) BARD1 (UniProt: Q99728) from Cluster 4.

**Table 1 molecules-27-00423-t001:** Top 10 clusters obtained using CD-HIT and their domain information.

Cluster No.	No. of Proteins	Average Copy No.	Std. Dev. Copy No.	InterPro Domains
1	34	10.9	8.1	Ankyrin repeat-containing domain, ZU5 domain, Death domain, EGF-like domain, Notch domain, Sterile alpha motif domain
2	32	5.2	0.8	Ankyrn repeat-containing domain, CCDC144C-like coiled-coil domain
3	13	10.7	6.8	Ankyrin repeat-containing domain, K Homology domain, Sterile alpha motif domain, Protein kinase domain
4	9	3.7	1	Ankyrin repeat-containing domain, BRCT domain, BCL-6 corepressor, PCGF1 binding domain
5	8	6.4	0.7	Ankyrin repeat-containing domain, NFkappaB IPT domain, Rel homology domain (RHD), Death domain
6	7	9.1	2.4	Ankyrin repeat-containing domain, SOCS box domain
7	6	6.0	2.4	Ankyrin repeat-containing domain
8	6	7.8	4.4	Ankyrin repeat-containing domain
9	6	5.0	1.5	Ankyrin repeat-containing domain, Ion transport domain
10	6	4.0	0	Ankyrin repeat-containing domain, Transient receptor ion channel domain, Ion transport domain

**Table 2 molecules-27-00423-t002:** Top 10 sequence clusters and their consensus sequence. The conserved positions are highlighted in blue color.

Cluster No.	Consensus (50%)
1	X**G**X**TPLH**X**AA**XX**G**XXXXXXX**LL**XX**GA**XXXX
2	XX**RTAL**XX**AV**XXXXXXX**V**XX**LL**XXXXXXXX
3	X**G**X**T**X**L**XX**A**XXX**GH**XXX**V**XX**LL**XX**GA**XXXX
4	X**G**X**TPLH**X**A**XXX**G**XXXXXXX**LL**XX**GA**XXXX
5	X**G**X**T**X**LHLA**XXXXXXXXXXX**L**XXX**G**XXXXX
6	XXXX**PL**XX**A**XXX**G**XXXXXXX**LL**XX**GA**XXXX
7	X**G**XXX**LH**X**AA**XX**G**XXXXXXX**L**XX**AG**XXXXX
8	X**G**X**T**X**LH**X**AA**XX**G**XXXXXXX**LL**XXX**A**XXXX
9	X**G**X**T**X**L**XX**A**XXXXXXXXXXX**L**XXXXXXXXX
10	XXXX**A**X**LLA**XXXXXXXX**V**XX**LL**XXXXXXXX

**Table 3 molecules-27-00423-t003:** Variations associated with the most conserved positions of the ankyrin motif.

Residue Position	Gene Name	Copy No.	Cluster No.	Variation	Condition(s)
Gly2	ANKRD29	4	1	G112E	G -> E (in dbSNP:rs17855552)
	RNASEL	2	3	G59S	G -> S (in dbSNP:rs151296858)
	BARD1	1	4	G428*	Hereditary cancer-predisposing syndrome
	CDKN2B	2	7	G47E	Lung adenocarcinoma
Thr4	ANKK1	8	1	T595I	T -> I (in dbSNP:rs55787008)
Pro5	ANKK1	8	1	P596L	P -> L (in dbSNP:rs7104979)
	ASB2	4	6	P160S	P -> S (in dbSNP:rs2295213)
	CDKN2A	4	7	P114L	Non-small cell lung carcinoma
	CDKN2A	4	7	P114S	Melanoma; lossof CDK4 binding; dbSNP:rs104894104
	CDKN2A	2	7	P48L	CMM2 ^$^; also found in head and neck tumor
	CDKN2A	2	7	P48T	Hereditary melanoma|Hereditary cancer-predisposing syndrome
	CDKN2A	3	7	P81L	Melanoma; impairs the function;
	CDKN2A	3	7	P81T	CMM2; loss of CDK4 binding;
Leu6	ANK1	8	1	L276R	Spherocytosis type 1
	ANK1	17	1	L573fs	Spherocytosis type 1
	ANKK1	1	1	L366F	L -> F (in dbSNP:rs56339158)
	NFKBIA	2	5	115..120	LHLAVI->AHAAVA: Greatly reduced nuclearlocalization. Great reduction in its ability to inhibit DNA binding of RELA.
	CDKN2A	1	7	L16fs	Hereditary cancer-predisposing syndrome
	CDKN2A	1	7	L16fs	Squamous cell lung carcinoma; Hereditary melanoma; Hereditary cancer-predisposing syndrome
	CDKN2A	1	7	L16P	Biliary tract tumor; familial melanoma
	CDKN2A	1	7	L16P	Hereditary melanoma
	CDKN2A	1	7	L16R	Hereditary melanoma; Hereditary cancer-predisposing syndrome
	INVS	15	8	L493S	NPHP2; impairs ability to target DVL1 fordegradation
His7	ANKK1	1	1	H367Q	H -> Q (in dbSNP:rs34298987)
	CDKN2A	3	7	H83N	Lung tumor
	CDKN2A	3	7	H83Q	H -> Q (in dbSNP:rs34968276)
	CDKN2A	3	7	H83Y	Pancreas tumor; head andneck tumor
Ala9	CDKN2A	1	7	A19T	CMM2; loss of CDK4 binding
	CDKN2A	4	7	A118T	CMM2
	CDKN2A	3	7	A85T	A -> T (in dbSNP:rs878853646)
Gly13	BARD1	3	4	G505fs	Hereditary cancer-predisposing syndrome; Familial cancer of breast
	CDKN2A	4	7	G122R	CMM2
	CDKN2A	4	7	G122S	Biliary tract tumor
	CDKN2A	1	7	G23D	Pancreas tumor; melanoma; loss of CDK4 binding
	CDKN2A	3	7	G89D	CMM2
	CDKN2A	3	7	G89S	CMM2
	TRPC6	1	10	G109S	Focal segmental glomerulosclerosis 2 (FSGS2); increases calcium ion transport
Leu21	CDKN2A	3	7	L97R	CMM2; loss of CDK4 binding
Leu22	CDKN2A	1	7	L32P	Hereditary cancer-predisposing syndrome; Hereditary melanoma
Gly25	ANK2	20	1	G685E	breast cancer
	ANKK1	3	1	G451R	G -> R (in dbSNP:rs34983219)
	ANKHD1	1	3	G228C	G -> C (in dbSNP:rs17850572)
	CDKN2A	3	7	G101W	Hereditary melanoma; Cutaneous malignant melanoma 2; Melanoma-pancreatic cancer syndrome; Hereditary cancer-predisposing syndrome
	CDKN2A	1	7	G35A	CMM2; biliary tract tumor; uveal melanoma; partial loss of CDK4 binding
	CDKN2A	1	7	G35E	CMM2
	CDKN2A	1	7	G35V	CMM2; loss of CDK4 binding
Ala26	ANKRD16	3	6	A128G	A -> G (in dbSNP:rs2296136)
	CDKN2A	3	7	A102E	Seminoma; medulloblastoma tissues from Li-Fraumeni syndrome patients carrying a mutation in TP53;
	CDKN2A	3	7	A102T	A -> T (in dbSNP:rs35741010)
	CDKN2A	1	7	A36fs	Hereditary melanoma; Hereditary cancer-predisposing syndrome

^$^ Cutaneous malignant melanoma 2.

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
