# Peer review of "Sequence and Structure-Based Analyses of Human Ankyrin Repeats"

_molecules, 2022, doi:10.3390/molecules27020423_

Round 1

Reviewer 1 Report

The paper is well written and clear. However, it lacks some details and analysis of the effect of the mutations on the structure of the ANK repeats. The authors have found structures and created theoretical models but no structural 3D analysis was performed. Structural analysis will improve the paper.

p. 11, In 'Discussion'. Elaborate more on 'non-ankyrin domains that affect the orientation of the ANK domain in the overall 3-dimensional structure of the protein' Provide some examples and figure(s).

Minor comments:

1. Provide an explanation of the MSA abbreviation (it only appears in Fig. 3)

2. p. 3, 2nd paragraph 'are and multiple sequence alignment performed...' need correction 

Reviewer 2 Report

This is a useful and carefully written study in which the authors have classified the human ankyrin repeat proteins into clusters based on the sequence similarity in their ankyrin repeat domains. They analyzed conserved and structurally important residues in the ankyrin repeat motifs and discuss their role in human diseases.

Although the ankyrin repeat domains are among the most common, and therefore best defined/described conserved structural motifs, this work is particularly valuable because it extends this knowledge by using the latest findings in structural biology, including deep learning methods. In addition, the data are correlated to the disease associated mutations.

In my opinion, the conclusions are consistent with the evidence presented.

I have no additional comments on the tables and figures.

In my opinion, the references are appropriate.

To answer the questions posed by the authors, I consider the bioinformatic methodology used to be correct and appropriate

I have only two minor recommendations for correction that I would like the authors to consider:

- Page 3, 2nd Paragraph, abbreviation for MSA (multiple sequence alignment)

- Page 8, last line, 3D instead of 3d

Reviewer 3 Report

The stydy of Chakrabarty and Perekh uses the latest structural and clinical data and performs sequence- and structure-based analyses of ankyrin motifs to decipher the role of conserved and structurally important residues. The strength of the study is the thoroughly conducted and described results section, which is easy to follow. This reviewer lacks a concluding paragraph at the end of the results section that would summarizes the new findings from this study and highlights how and why they are important for a better understanding of the structure-function relationship of ankyrin repeats - it is not clear. Another strong point of the study is Table 3, where the authors mined the ClinVar database for pathological variations associated with the most conserved positions of ankyrin motif. Again, this reviewer also lacks a concluding paragraph in the discussion section that could provide some concluding remarks regarding commonalities or some general principle, if any, that could be built upon their results. The weakness of the study is its very descriptive nature without a clearly defined objective. The authors have performed a careful analysis, which they use in their discussion of disease-associated mutations. However, the clearly stated conclusion of the study is missing.

Minor comments:

  1. Page 3, line: 25 are part of the two-residue turns T1, T2 and T2 respectively of the ankyrin motif, as

The second T2 should be T3?

  1. In my pdf version Figure 3 is included in Table 1
  2. The article is not consistent in marking 3-dimensional, somewhere 3D, somewhere 3d (last sentence of page 8), somewhere 3-dimensional.
  3. The article is not consistent in marking mutations somewhere N125S, somewhere Gly2Ser (page 14)

Round 2

Reviewer 1 Report

I'm satisfied with the revision.